# Climate variation and serotype competition drive dengue outbreak dynamics in Singapore

Emilie Finch [1,2] ✉, Chia-chen Chang[3,4], Adam Kucharski[1], Shuzhen Sim[3], Lee-Ching Ng [3,5,6] & Rachel Lowe [1,6,7,8,9] ✉

Dengue poses a rapidly increasing threat to global health, with Southeast Asia among the most affected regions. Climate-informed early warning systems can help mitigate outbreaks; however, prediction of large outbreaks with sufficient lead time remains a challenge. In this study, we quantified the role of climatic variation and serotype competition in shaping dengue risk in Singapore using over 20 years of weekly case data. We integrated these insights into a forecasting framework capable of predicting dengue outbreaks up to two months ahead and generated counterfactual projections to assess the impact of novel interventions, such as *Wolbachia*. While a climate-informed model improved predictive power by 54% compared to a seasonal baseline, including serotype information increased this to 60%, better explaining interannual variation in dengue incidence. By incorporating serotype competition as a proxy for population immunity, this work advances the field of climate-informed dengue prediction and demonstrates the value of long-term virus surveillance.

Climate-informed forecasting models can provide advance warning of infectious disease outbreaks and help mitigate their impact. As the climate crisis drives more frequent extreme weather events and alters the dynamics of climate-sensitive diseases such as dengue, early warning systems that integrate climate information are becoming an increasingly important tool for epidemic preparedness and response[1,2].

Dengue is an emerging vector-borne disease, transmitted by *Aedes aegypti* and *Aedes albopictus* in urban and peri-urban areas[3]. Global reported dengue incidence has increased 30-fold over the past 50 years, accompanied by an expansion in the geographical range of transmission, and approximately half of the global population is now thought to be at risk of dengue infection[4]. There are four antigenically distinct serotypes of dengue virus (DENV1-4): infection with one serotype results in life-long immunity to the infecting serotype, and

temporary cross-immunity to others[5,6]. Dengue epidemic dynamics are complex and challenging to predict, with large outbreaks driven by multiple factors including climate variation, competition between the four dengue serotypes, prior exposure to dengue in the population, and traditional and novel vector control efforts.

Singapore, an equatorial city-state in Southeast Asia, experiences hyperendemic dengue transmission with all four serotypes in circulation and cyclical replacements in the dominant serotype[7,8]. Singapore experiences warm and humid temperatures year round, with suitable conditions for mosquito breeding and dengue transmission. Rainfall is affected by two monsoon seasons, with the Northeast monsoon occurring from December until early March and the Southwest monsoon occurring from June to September[9]. Since the 1960s, Singapore has implemented stringent dengue prevention measures focused on vector control and public education. This has led to a reduction in the

[1]Centre for Mathematical Modelling of Infectious Diseases, London School of Hygiene and Tropical Medicine, London, UK. [2]Department of Genetics, University of Cambridge, Cambridge, UK. [3]National Environment Agency, Singapore, Singapore. [4]Department of Biological Sciences, National University of Singapore, Singapore, Singapore. [5]School of Biological Sciences, Nanyang Technological University, Singapore, Singapore. [6]Saw Swee Hock School of Public Health, National University of Singapore, Singapore, Singapore. [7]Centre on Climate Change and Planetary Health, London School of Hygiene and Tropical Medicine, London, UK. [8]Barcelona Supercomputing Center (BSC), Barcelona, Spain. [9]Catalan Institution for Research and Advanced Studies (ICREA), Barcelona, Spain. ✉e-mail: ef507@cam.ac.uk; rachel.lowe@bsc.es

*Aedes* House Index (a measure of the percentage of houses positive for *Aedes* breeding) from 48% in 1966 to around 1% in the 1990s[10]. Periodic seroprevalence surveys have demonstrated a concurrent decrease in seroprevalence in almost all age groups, with an estimated decrease in force of infection from around 0.1 per year in the 1960s to 0.01 per year from the 1990s onwards[11,12]. Despite this, reported cases have increased in recent years. Possible explanations for this include improved case detection and reporting, particularly after 2008, where a campaign was launched encouraging the use of non-structural protein 1 (NS1) antigen rapid tests in laboratories, or higher population vulnerability to dengue outbreaks (for example, after the importation of a new viral genotype) due to lower immunity or prior dengue exposure[7,10,12].

From 2016 onwards, the National Environment Agency of Singapore (NEA) has conducted phased field trials to investigate augmenting traditional vector control with a *Wolbachia-Aedes* mosquito suppression strategy, Project *Wolbachia*[13,14]. This employs *Wolbachia*-based incompatible insect technique (IIT), where male mosquitoes infected with *Wolbachia*, a maternally inherited intracellular bacterium, are released. As mating between *Wolbachia*-infected males and wildtype females results in non-viable offspring due to cytoplasmic incompatibility, these releases are expected to suppress mosquito populations and reduce dengue transmission[15]. This is then coupled with sterile insect technique (IIT-SIT), where irradiation is used to sterilise any residual females in releases of *Wolbachia*-infected males, to reduce the risk of stable establishment in the mosquito population. Analysis of large-scale field trials conducted in Singapore from 2018 to 2022 showed a reduction in dengue incidence rates in intervention locations when compared with synthetic controls, estimating an intervention efficacy of 77.3%[14]. In 2022, a multi-site cluster randomised controlled trial was implemented, which further expanded release coverage in Singapore, with trial results forthcoming[13].

Climate influences dengue transmission through effects on the vector (*Aedes* mosquitoes) and the dengue virus itself. Temperature affects mosquito survival, development and reproduction, as well as the viral extrinsic incubation period, with an optimal temperature for transmission of around 29 °C and thermal limits between 17.8 °C and 34.5 °C[16]. Temperature has been found to shape the timing, length and geographical extent of dengue seasons[2,17]. Contrastingly, the impact of rainfall on dengue transmission is more nuanced. While increasingly wet and humid conditions can lead to the creation of mosquito breeding sites, the effect of rainfall on transmission is dependent on human water storage behaviour, and the availability of water and sanitation infrastructure, which can lead to non-linear and delayed impacts of rainfall on dengue transmission[18–20]. In particular, excessive rainfall can lead to flushing effects, where mosquito breeding habitats are washed away entirely. This has been documented in Singapore, where dengue outbreak risk was found to decrease following flushing events[21]. Previous research in Singapore has demonstrated the utility of temperature, precipitation and absolute humidity in predicting dengue incidence[22–24]. Dengue outbreaks have also been associated with El Niño events, the warm phase of the El Niño Southern Oscillation (ENSO), involving warmer than normal oceanic and atmospheric temperatures in the Pacific, which typically lead to hotter and drier climatic conditions in Singapore[25–27]. Currently, Singapore uses a machine learning approach based on Least Absolute Shrinkage and Selection Operator (LASSO) regression to generate operational forecasts of dengue incidence for outbreak alerts and decision-support[28].

Forecasting models can capitalise on inherent lags between climatic variation and dengue transmission to predict outbreak risk at operationally useful lead times[29]. In Singapore, an analysis of vector control found that local authorities needed an average of 2 months to mitigate the impact of a dengue outbreak, suggesting that early warning forecasts with several months lead time would be optimal[30]. Forecasts at shorter horizons may also be helpful to inform situational awareness. To date, dengue forecasting models have struggled to predict interannual variability in dengue seasons and shown worse predictive performance for high incidence seasons, which have the greatest public health impact[31]. Additionally, forecast skill is typically lower earlier in the season when aiming to forecast several months ahead, which is when forecasts have the most potential operational value. While immunity is theoretically recognised as an important driver of interannual dengue dynamics, to our knowledge, no current dengue forecasting models directly account for serotype dynamics or changes in immunity, which are likely to be particularly important in hyperendemic regions such as Southeast Asia[31–34]. To address this, we incorporated climate and serotype dynamics within a Bayesian hierarchical modelling framework to forecast dengue incidence. We quantified the effect of climatic variables, the Niño 3.4 index and switches in dominant serotype on dengue incidence, and evaluated forecasts of dengue cases with a 2–8 week forecast horizon. Finally, we demonstrate two real-world applications of this framework; first, to produce operational dengue forecasts in Singapore and second, to explore the impact of novel dengue interventions by generating counterfactual scenarios.

## Results

### Reported dengue cases in Singapore over the past two decades

Between 1 January 2000 and 31 December 2022, 234,358 cases of dengue were reported in Singapore. Most cases were reported between June and September, typically following warm and humid climatic conditions (Fig. 1, panels D–F). Cyclical outbreaks, which were historically thought to occur every 5–6 years, have become more frequent and of larger magnitude (Fig. 1). For instance, while the 2004 dengue outbreak led to 9459 cases overall and peaked at 332 reported weekly cases, the 2022 outbreak resulted in 32,259 reported cases and peaked at 1568 reported weekly cases. We defined an outbreak week using a seasonal moving 75th percentile threshold.

Major outbreaks (where over a third of the year was defined as an outbreak week) were identified in; 2004, 2005, 2007, 2008, 2013, 2014, 2015, 2016, 2019, 2020 and 2022. In some years, dengue outbreaks coincided with switches in dominant serotype, such as the switch from DENV-2 to DENV-1 in 2013 or from DENV-2 to DENV-3 in 2022 (Fig. 1). Additionally, outbreaks sometimes coincided with El Niño events, defined when sea surface temperature anomalies (SSTAs) in the Niño 3.4 region exceed 0.5 °C for 6 consecutive months.

### Climate and serotype dynamics shape dengue risk with non-linear and delayed effects

To quantify the effect of climate, ENSO, and changes in serotype on dengue risk, we fit a Bayesian hierarchical model to weekly case data. We used a negative binomial likelihood and incorporated weekly random effects to capture seasonality and yearly random effects to account for unexplained interannual variation in dengue risk, for instance, due to changes in vector control intensity ("Methods"). We then compared a model including only weekly and yearly random effects with models containing additional climatic and serotype covariates to see whether their inclusion improved model adequacy statistics, and assessed the extent to which covariates accounted for interannual variation in the model. We tested temperature, precipitation, humidity and ENSO variables considering lags from 0 to 20 weeks, as well as serotype variables (Supplementary Table 1). We selected a final model including: 12-week rolling average maximum temperature in °C; 12-week total days without rain; 12-week rolling average Niño 3.4 SSTA with a 4 week lag; and a time-varying covariate measuring the number of weeks since a switch in dominant serotype ("Methods"). We found a non-linear relationship between maximum temperature and dengue incidence risk, with increased risk around 32 °C and decreased risk at particularly low or high maximum temperatures (Fig. 2, panel A). Similarly, we found increased risk of dengue

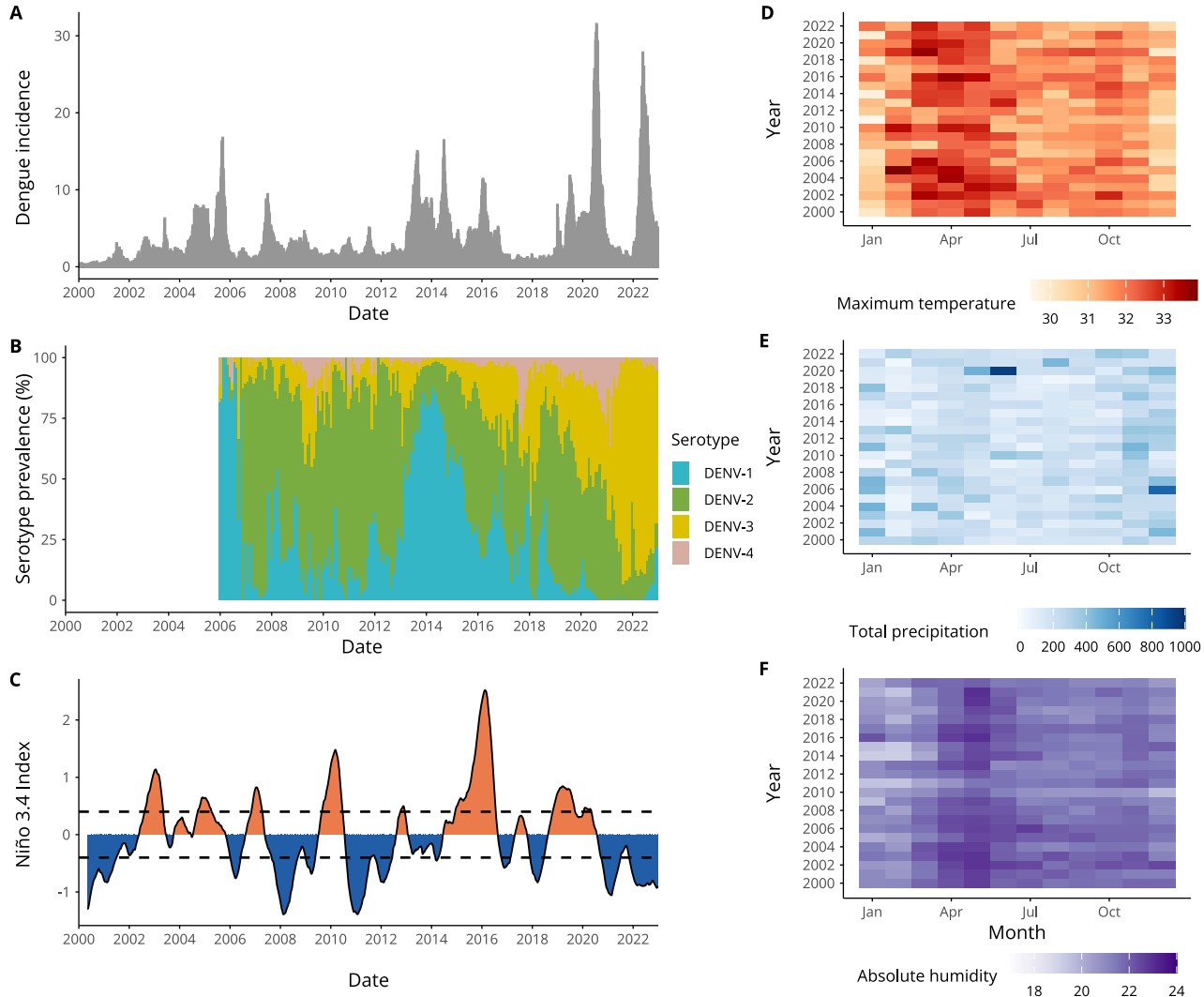

**Fig. 1 | Dengue cases, climate variability and serotype dominance in Singapore.** Figure showing over two decades of epidemiological and climatic data for Singapore. **A** Bars show weekly dengue incidence per 100,000 population from 1 January 2000 to 31 December 2022. **B** Stacked bars show the proportion of each serotype detected through virus surveillance. To calculate this, we aggregate the number of serotyped cases for DENV1-4 at a monthly level and divide each by the total number of cases serotyped in that month. **C** Line graph showing weekly Niño 3.4 sea surface temperature anomalies (SSTAs), dashed lines indicate +0.5 °C and −0.5 °C, which are often used as thresholds to define El Niño and La Niña events. Blue shading indicates negative values of the Niño 3.4 index (indicating La Niña conditions), while orange shading indicates positive values, associated with El Niño conditions. Heatmaps for (**D**) monthly mean maximum temperature (°C), **E** total precipitation (mm), and **F** mean absolute humidity (g/m³).

at intermediately wet conditions, with around 30 days without rain in the previous 3 months, and decreased risk in dry conditions, with more than 45 days without rain in the previous 3 months (Fig. 2, panel B). We found decreased dengue risk with negative Niño 3.4 SSTA values and non-linearly increasing dengue risk with values of Niño 3.4 SSTA upwards of around 1.4 (Fig. 2, panel C). Finally, we found a non-linear relationship between the time since a switch in dominant serotype and dengue risk, with increased risk in the first two years following a switch, decreased risk between 2 and 6 years following a switch, and subsequent increased risk at 6+ years following a switch (Fig. 2, panel D).

We then compared the yearly random effects estimated for our final *climate and serotype* model, a *climate only* model including climatic covariates and random effects, a *serotype only* model including the serotype covariate and random effects, and a *random effects only* model including weekly and yearly random effects (Fig. 2, panel E). As yearly random effects account for unexplained interannual variation in

dengue incidence, we would expect estimated yearly values to be closer to 0 when other model covariates are able to explain this variation. We found no difference in estimated yearly random effects between the four models before 2006, which we would expect, as no serotype information is available before this date. From 2007 onwards, overall, models including serotype information tended to have yearly random effects closer to 0 when compared to a *climate only* model or *random effects only* model. We quantified the contribution of covariates in accounting for interannual variation in dengue incidence by calculating the ratio of marginal variances of the covariate model and the *random effects only* model (Methods). While a *climate only* model did not reduce the variance in the yearly random effect (with a ratio of 1.01), including additional serotype information resulted in a ratio of 0.68, suggesting these covariates are able to capture some of the interannual variation in dengue incidence. Similarly, a *serotype only* model showed reduced variance in the yearly random effect with a ratio of marginal variances of 0.71.

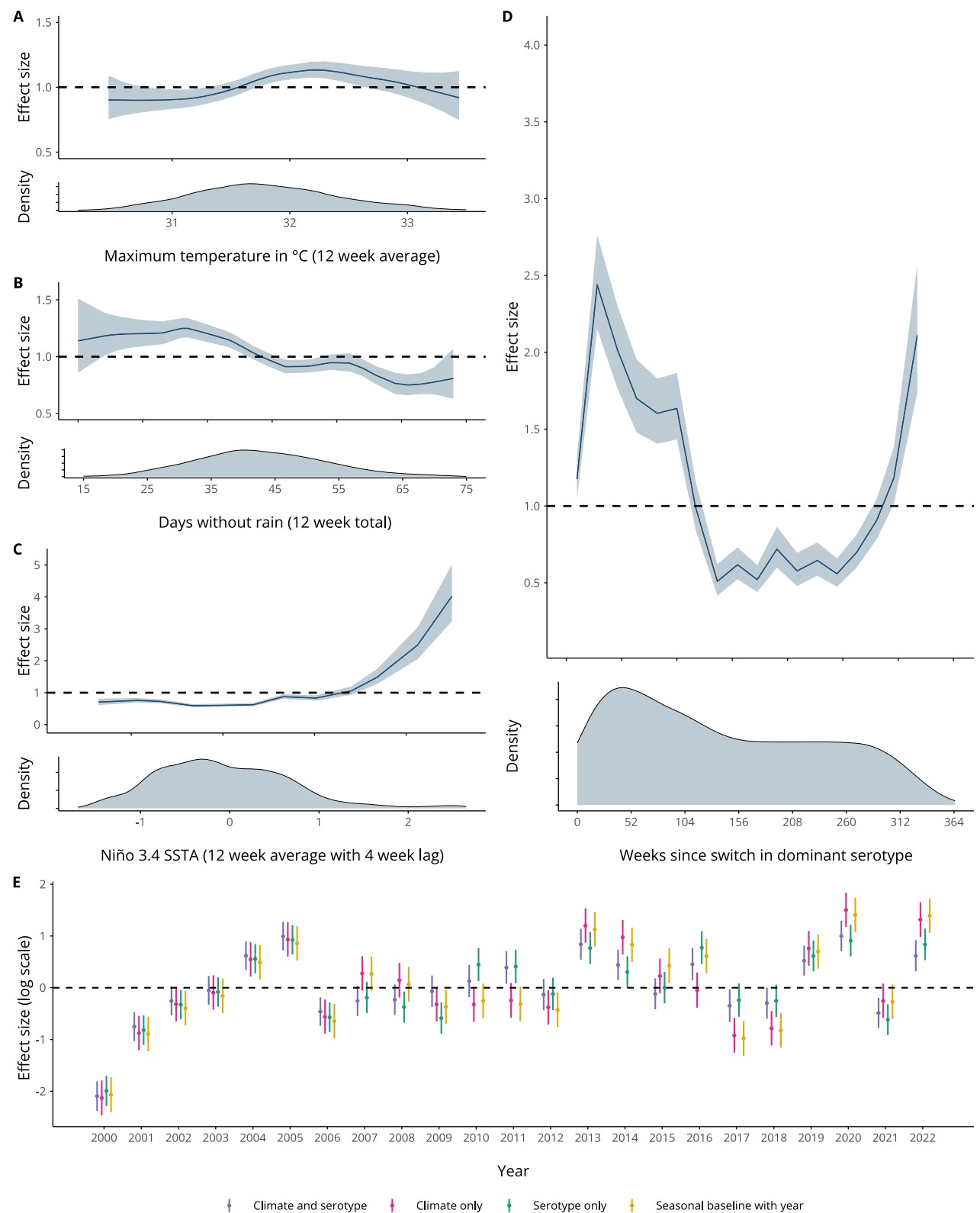

## Accounting for serotype and climate dynamics improves probabilistic predictions of dengue case incidence and outbreak detection

To assess the predictive ability of the identified relationships between climate, serotype changes and dengue incidence, we used a time series cross-validation approach to produce probabilistic dengue predictions and calculate the probability of exceeding a predefined outbreak threshold in a given week[35]. We first generated probabilistic predictions with no lead time (using information up to a target date to predict dengue cases on that date) to validate our candidate models and compare out-of-sample predictive ability between 2009 and 2022 (Fig. 3). The first 8 years of data were used exclusively for training ("Methods"). We compared our final selected *climate and serotype* model with a *climate only* model and a *serotype only* model containing only climatic and serotype covariates, respectively (Supplementary Table 3). We compared these to a *seasonal baseline* model, which only

**Fig. 2 | Effects of climatic variability and switches in dominant serotype on dengue incidence in Singapore. A–D** show posterior marginal effects and density plots for covariates in the final selected model. These include maximum temperature in °C (12 week running average), days without rain (12 week total), Niño 3.4 SSTA (12 week average with a 4 week lag) and weeks since switch in dominant serotype. These are shown on the relative risk scale displaying the median value and associated 95% credible interval and can be interpreted as the effect of the covariate on dengue incidence rate with all other parameters held constant. **E** compares the estimated mean yearly random effect, $\gamma_{a[t]}$, and associated 95% credible interval for a *random effects only* model including only weekly and yearly random effects $\gamma_{a[t]} + \delta_{w[t]}$ (in yellow), the final selected *climate and serotype* model including all climate and serotype covariates and random effects (in purple), a *climate only model* with random effects (in pink), and a *serotype only* model with random effects (in green). The estimated yearly random effect from the *random effects only* model indicates whether dengue incidence was higher or lower for that year than the overall mean incidence. We would expect the estimated yearly random effects for covariate models to be closer to 0 (indicated with a dashed line) when covariates are able to account for interannual variability in dengue incidence. These estimates are based on 1201 observations of reported dengue cases, as shown in Fig. 1.

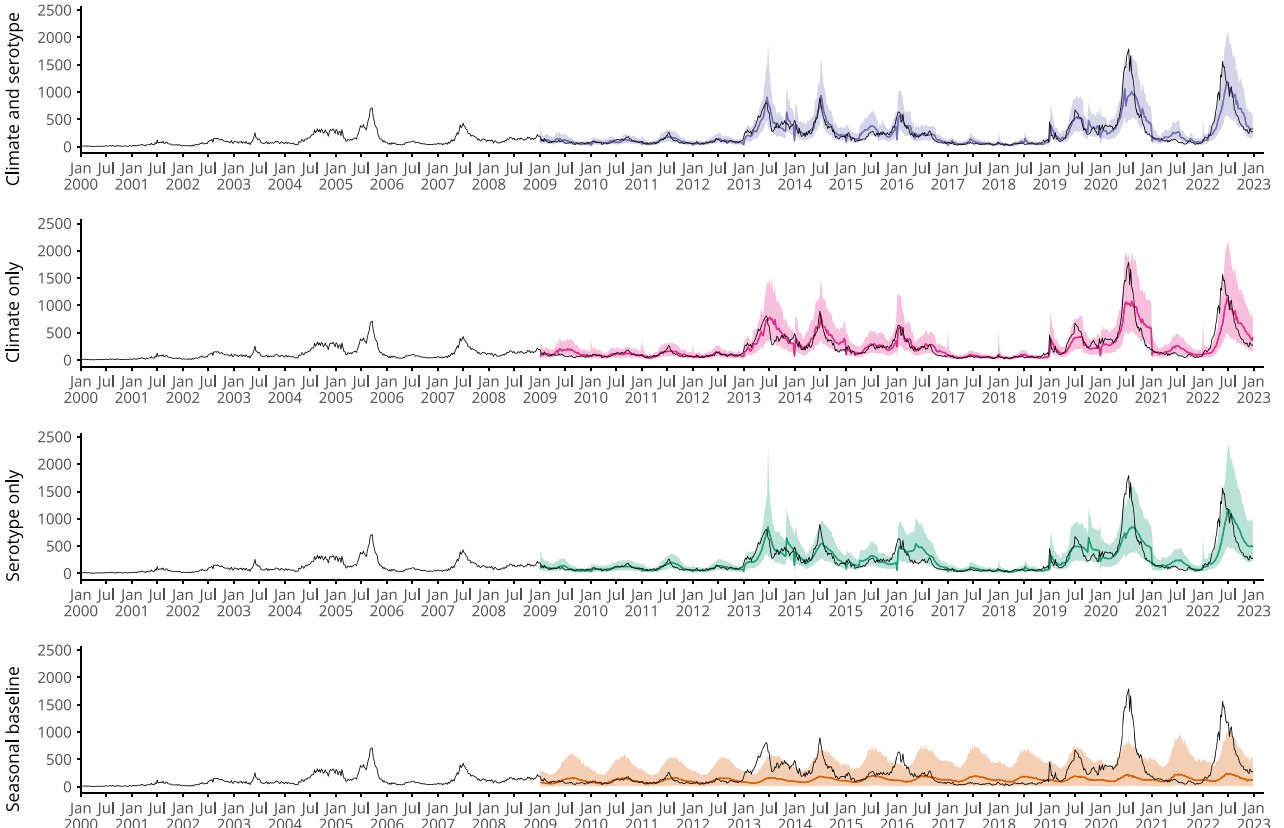

**Fig. 3 | Comparing time series cross-validated predictions of candidate models.** Figure showing time series cross-validated posterior predictions of dengue cases for each model from 2009 to 2022. We used an expanding window cross-validation methodology, where the model is trained on data up to but not including the target week and then posterior predictions are generated for the target week. Coloured lines show the median posterior prediction of weekly dengue cases, shaded areas show the 95% prediction interval and the dark grey lines show the data. From top to bottom the figure shows: predictions for the final selected *climate and serotype* model with weekly and yearly random effects $\gamma_{a[t]} + \delta_{w[t]}$ in purple; predictions for a *climate only* model with weekly and yearly random effects in pink; predictions for a *serotype only* model with weekly and yearly random effects in green; and predictions from a *seasonal baseline* model with only weekly random effects $\delta_{w[t]}$ in orange.

included weekly random effects. This is equivalent to a naive baseline model, which uses the historical seasonal pattern in dengue incidence to predict cases in a given target week. This is similar to the endemic channel threshold used operationally in Singapore, which is calculated using cases reported in previous years to monitor outbreaks in the current year (Supplementary Fig. 1). We assessed forecast skill in predicting weekly dengue cases using the continuous ranked probability score (CRPS), where smaller values indicate better performance. We also calculated the continuous ranked probability skill score (CRPSS), which is defined as the percentage improvement in CRPS compared to the *seasonal baseline* model. We assessed the predictive ability of the candidate models for outbreak detection using the Brier score and conducted receiver operating characteristic (ROC) analysis. Here, we compared the model hit rate (proportion of outbreak weeks correctly

identified) with the false alarm rate (the proportion of weeks without an outbreak where an outbreak was predicted to occur). We calculated the area under the curve (AUC) to measure model skill in classifying outbreak and non-outbreak weeks ("Methods").

The *climate and serotype* model was able to reproduce dengue epidemic dynamics in Singapore between 2009 and 2022 and had a lower CRPS than all other models, showing a 60% relative improvement over the *seasonal baseline* model according to the CRPSS (Fig. 3). The *climate only* and *serotype only* models also performed well, with a 54% and 49% relative improvement over the *seasonal baseline* model respectively (Supplementary Table 3). In particular, the *climate and serotype* model is better able to predict the decrease in early 2016 than a *serotype only* model, which predicts a late peak around July 2016. Similarly, the *climate and serotype* model outperforms the *climate only*

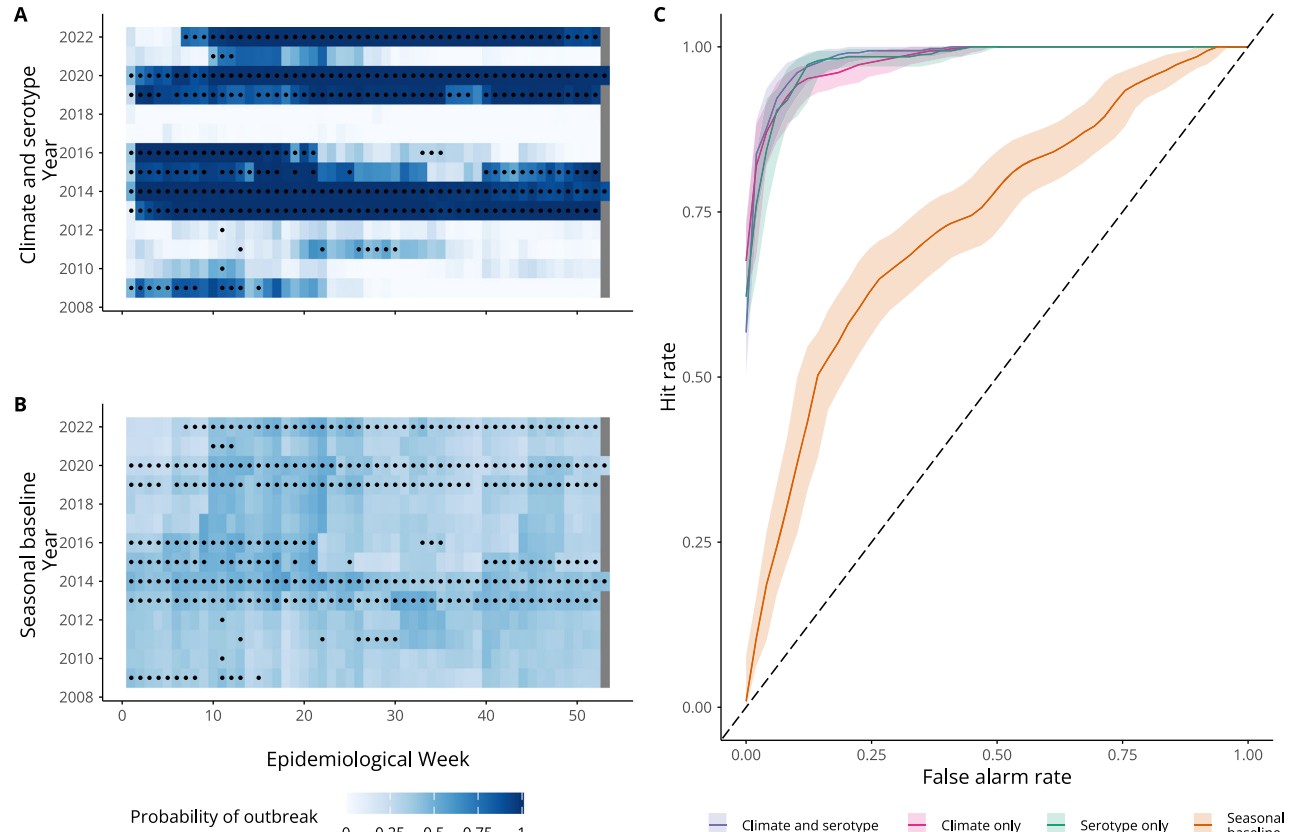

**Fig. 4 | Comparing outbreak detection of candidate models. A, B** show tile plots of model posterior predictions of exceeding the outbreak threshold for each week between 2009 and 2022. White indicates $P_{outbreak} = 0$, while dark blue indicates $P_{outbreak} = 1$. Circles indicate observed outbreak weeks defined using a seasonal moving 75th percentile threshold. For a given month and year, we defined an outbreak threshold using the 75th percentile of weekly cases in that month using all years up to, but not including, the given year. **C** shows an ROC curve, with the

coloured line showing hit rate and false alarm rate for different model outbreak alert thresholds. Hit rate (or sensitivity) is defined as the proportion of outbreak weeks that were correctly predicted. False alarm rate (1−specificity) is defined as the proportion of weeks without an outbreak where an outbreak was predicted to occur. The shaded area shows the 95% confidence interval of the sensitivity (hit rate) for a given false alarm rate.

model in predicting the decrease in cases following peaks in mid-2013 and mid-2014, as well as better predicting peak cases in July 2019. All covariate models underpredicted the peak in 2020 but were able to accurately predict peak timing. Similarly, in 2022, covariate models underpredicted the peak and predicted later peak timing than what was seen. The *climate and serotype* model outperformed other models in outbreak detection, with a lower Brier score (indicating better performance). This can be seen in Fig. 4, where the *climate and serotype* model is better able to assign high probability of an outbreak to actual outbreak weeks and lower probability of an outbreak to non-outbreak weeks. The *climate and serotype* model also had the highest AUC (98%, 95% CI: 97.7–99.0%) and corresponding lowest false alarm rate (2.1%) and the highest hit rate (92%) of the candidate models, with an optimal model outbreak alert threshold of 71% (Supplementary Table 4). The *climate only* and *serotype only* models also performed well in outbreak detection, as can be seen from the overlapping ROC curves in Fig. 4.

**Dengue forecasting for early warning with 2−8 weeks lead time**
Having identified a model capable of reproducing dengue epidemic dynamics and identifying outbreak weeks, we then adapted our framework for use in an early warning context, producing forecasts with 2 to 8 weeks lead time. For each forecast horizon, we used the best approximation of the covariates in the final models available at the forecast issue date ("Methods", Supplementary Fig. 2). We then produced probabilistic predictions of dengue using our candidate models to compare predictive ability at 2–8 week forecast horizons. Forecasts for 4 weeks ahead and 8 weeks ahead are shown in Fig. 5, with forecasts

for all horizons in Supplementary Fig. 3. As expected, predictive ability declined with forecast horizon, with better performance at shorter lead times and increased uncertainty around model predictions at longer lead times (Fig. 6). For instance, at an 8 week forecast horizon, the *climate and serotype* model struggled to predict peaks in late 2015–2016, first over and then under-predicting. However, all three covariate models considered were able to capture broad epidemic dynamics from 2009 to 2022 even at longer lead times and offered considerable improvement in performance compared with the *seasonal baseline*. For instance, at an 8 week forecast horizon, the *climate and serotype* model showed 32% relative skill improvement over the *seasonal baseline*. Additionally, the *climate and serotype* model continued to accurately detect outbreak weeks at all forecast horizons, with an AUC of 94% (95% CI: 92.7–95.7%) at an 8 week forecast horizon (Supplementary Table 4).

We compared forecast skill metrics for each model from 0 to 8 weeks ahead and found the *climate and serotype* model had a lower CRPS than other models (indicating better performance) until a 6-week forecast horizon, when its performance was equivalent to the *climate only* model. When considering outbreak detection metrics (Brier score and AUC) the *climate and serotype* model outperformed all other models at all forecast horizons, with high AUC values also obtained by other candidate models (Fig. 6). We also calculated forecast skill by calendar month to compare predictive performance early and late in the dengue season and found similar performance between our candidate models from February onwards (Supplementary Fig. 4).

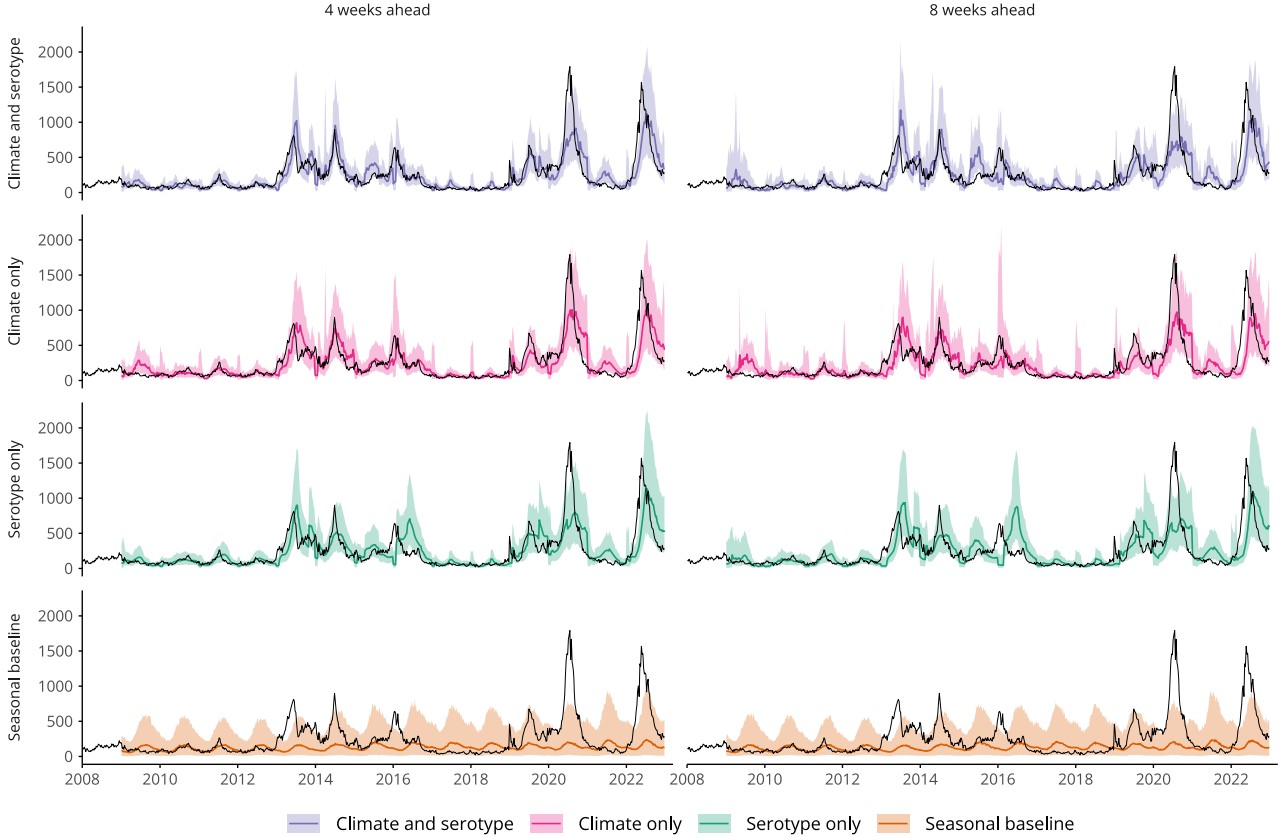

**Fig. 5 | Dengue forecasts for early warning at 4 and 8 week forecast horizons.** Figure showing time series cross-validated posterior predictions of dengue cases for each model from 2009 to 2022 at 2–8 week forecast horizons. We used an expanding window time series cross-validation methodology, where posterior predictions for each week are generated from a model fit to data up to week $t - h$, where $t$ is the target week and $h$ is the forecast horizon. Coloured lines show the median posterior prediction of weekly dengue cases, shaded areas show the 95% prediction interval and the dark grey line shows the data. From top to bottom the figure shows: predictions for the final selected *climate and serotype* model with weekly and yearly random effects $\gamma_{a[t]} + \delta_{w[t]}$ in purple; predictions for a *climate only* model with weekly and yearly random effects in pink; predictions for a *serotype only* model with weekly and yearly random effects in green; and predictions from a *seasonal baseline* model with only weekly random effects, $\delta_{w[t]}$, in orange. From left to right each column shows forecasts at 4 and 8 weeks ahead, respectively.

To optimise the proposed *climate and serotype* model for operational use, we then incorporated lagged cases as a predictor. This allowed us to compare 2–8 week ahead case forecasts for dengue seasons between 2019 and 2022 from this operational version of the proposed INLA model with forecasts generated by an updated version of a LASSO model developed by the NEA (Supplementary Fig. 7)[28]. Overall, we found similar predictive ability between our proposed INLA model and the LASSO model, with both outperforming a simple seasonal baseline (Supplementary Fig. 8). However, the models showed differing accuracy over time; while the LASSO model was better able to capture the 2020 peak in cases (particularly in 2–4 week ahead forecasts), the INLA model predicted the timing and decline of the 2022 peak in cases with greater accuracy (Supplementary Fig. 7).

### Estimating the impact of male *Wolbachia-Aedes* releases
Finally, we used our model framework to conduct scenario analysis, estimating the impact of Singapore's *Wolbachia-Aedes* suppression strategy on reported dengue cases in the 2023 dengue season. As the original *climate and serotype* model is able to predict dengue cases accurately without including lagged cases as a predictor, we were able to use climate and serotype patterns during the intervention period to estimate the cases averted by the expansion of *Wolbachia* releases. We trained the *climate and serotype* model with data until 26th June 2022. The training end date aligned with the beginning of a multi-site cluster randomised controlled trial of *Wolbachia* releases in Singapore rolled

out from July to September 2022, in which over 150,000 additional households were exposed to *Wolbachia* releases, corresponding to an 100% increase in household coverage[13,36]. We then used climatic and serotype data for 2023 to simulate expected weekly dengue cases in the absence of this expansion of *Wolbachia* releases (Fig. 7 and Supplementary Fig. 9). We estimated that 13,748 cases (95% PI: 9943–18,659) would have occurred in a counterfactual scenario where the expansion had not taken place, suggesting that the *Wolbachia* roll-out in this period averted around 3789 cases in 2023. This corresponds to 28% of the predicted cases for this period, according to the median predicted value (Supplementary Table 6). As *Wolbachia* releases were rolled out gradually in Singapore in several phases from 2018 to 2022, we conducted sensitivity analysis around the training end date, finding that earlier training end dates resulted in lower estimated numbers of cases averted in 2023 (Supplementary Table 6).

## Discussion
Climate and serotype dynamics have important impacts on dengue transmission and outbreak risk. However, to date, few statistical forecasting models account for both drivers within the same framework. This could lead to misattribution of dengue risk caused by changes in population immunity to climatic or other non-climatic factors, and limit the ability to forecast large outbreaks in advance. Additionally, mechanistic approaches aiming to better capture the biological processes underlying transmission have been found to

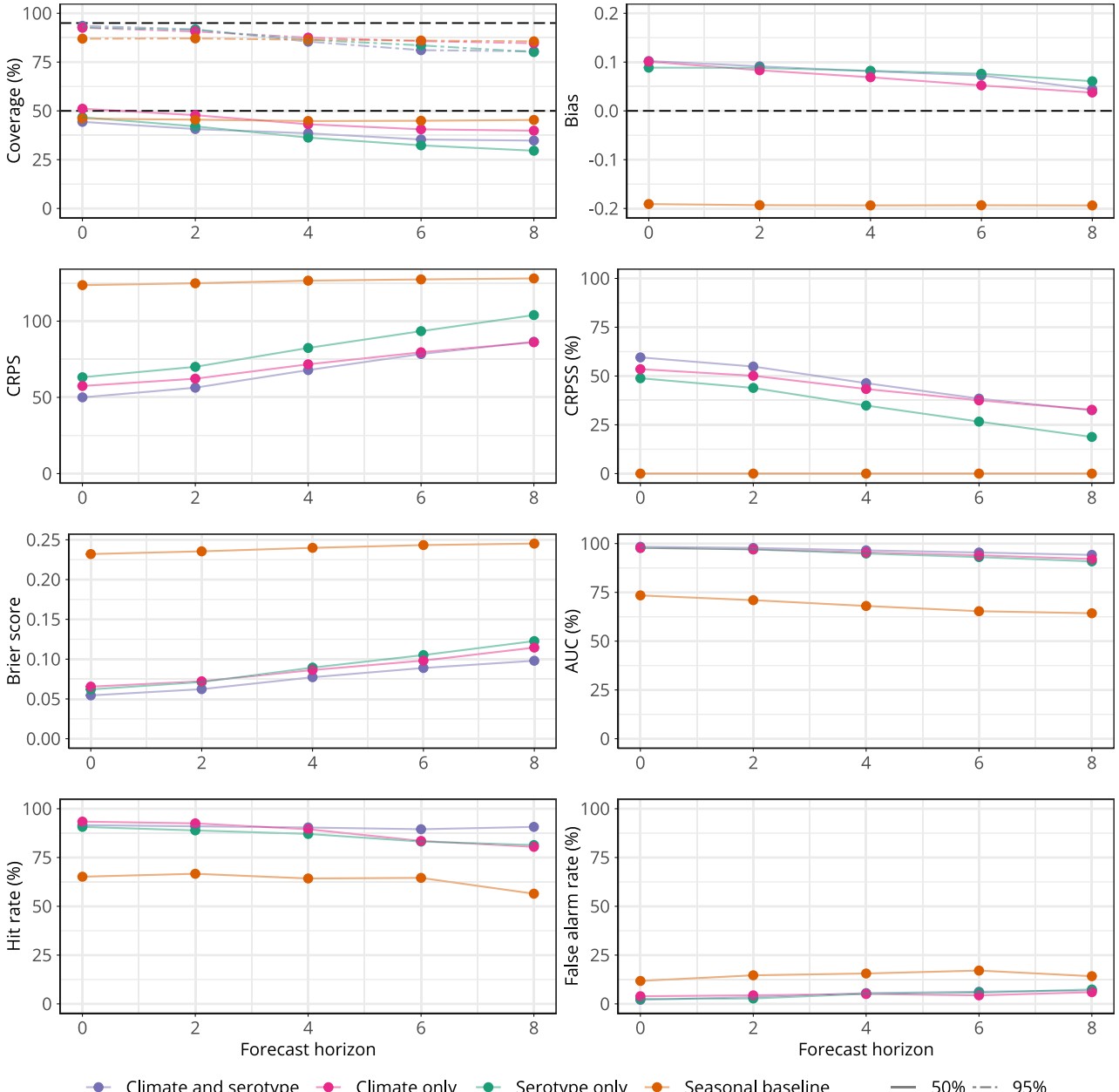

**Fig. 6 | Predictive performance over different forecast horizons.** Figure showing forecast metrics for each model across all forecast horizons from 2009 to 2022. From top left to bottom right these show: interval coverage %; bias; CRPS (continuous ranked probability score); CRPSS (continuous ranked probability skill score, %); Brier score; AUC (area under the curve, %); hit rate (%); and false alarm rate (%). Interval coverage shows the percentage of observations falling inside a given prediction interval. A perfectly calibrated forecast would have coverage equal to the nominal prediction interval; that is, 95% coverage equal to 95% and 50% coverage equal to 50%, indicated by dashed horizontal lines. Bias measures the relative tendency of the model to over- or under-predict, and is bounded between −1 and 1, with 0 indicating unbiased forecasts. The CRPS can take values between 0 and infinity, with smaller values indicating better performance. The CRPSS indicates the relative improvement of each covariate model over the *seasonal baseline* model and can take values from 0%, indicating that the model performs the same as the baseline, and 100%, indicating perfect forecasting skill. The Brier score can take values from 0 to 1, with smaller values indicating better performance. The AUC can take values from 0 to 100% with 100% indicating perfect classification. Hit rate and false alarm rate also take values from 0 to 100% with higher and lower values indicating better performance, respectively.

perform less well than statistical approaches to forecast dengue[31]. We analyse 20 years of data from Singapore to understand the relative impact of climatic and serotype dynamics on dengue risk and to produce probabilistic forecasts from 0 to 8 weeks ahead. Our approach integrates a proxy for changes in population immunity and the epidemic potential of the dominant virus in circulation within a statistical framework, aiming to generate accurate probabilistic predictions of dengue incidence through improved inference.

The impacts of climatic variables on dengue risk in Singapore were complex, with non-linear and delayed effects. We found increased risk of dengue at a maximum temperature of 32 °C and at intermediately wet conditions, with decreasing risk in very hot and very dry conditions. This suggests that dengue seasonality in Singapore may change as climate change leads to increasing temperatures, with fewer cases in the middle of the year. We also found non-linearly increasing dengue risk with increasing Niño 3.4 SSTAs, reflecting

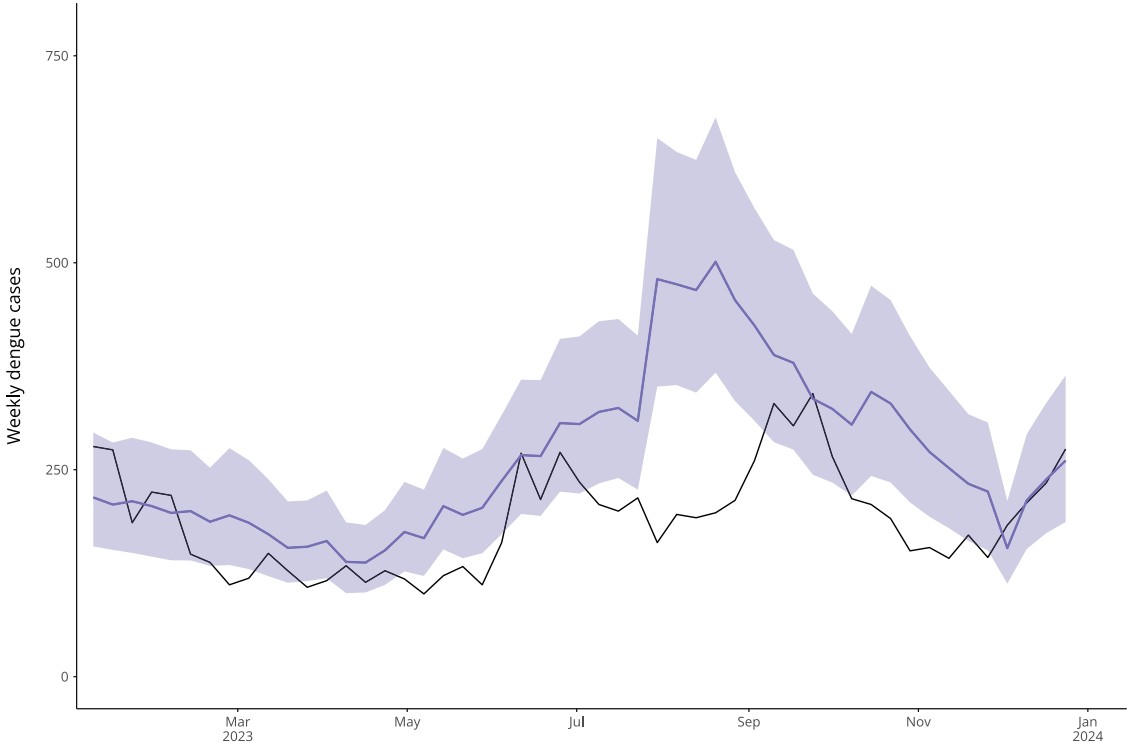

**Fig. 7 | Estimated impact of *Wolbachia* releases for 2023 dengue season.** Figure comparing predicted dengue cases using the *climate and serotype* model (which doesn't incorporate *Wolbachia* releases) with observed cases against a background of *Wolbachia* releases in 2023. The model was tuned using data until 26th June 2022 to align with the start of a randomised controlled trial of *Wolbachia* releases in Singapore. To estimate the impact of *Wolbachia* releases after this, out-of-sample predictions were generated from this point to the end of 2023 using the *climate and serotype* model. Here, the black line shows reported dengue cases while the purple line shows median model predictions for each week, and the shaded area shows the associated 95% prediction interval.

increased risk during El Niño conditions. We found a non-linear relationship between the time since a switch in dominant serotype and dengue transmission. In the first two years following a switch, when population immunity to the new serotype is low, we found increased dengue risk, followed by decreased dengue risk in the subsequent 4 years as immunity to the dominant serotype increases in the population. We then found evidence of increased risk at 6+ years following a serotype switch, which likely reflects the accumulation of susceptibles in the population as well as the risk associated with the growth of a non-dominant serotype before it reaches dominance, for instance, growing DENV-3 prevalence in 2019. Our results are in line with previous research on climate-dengue relationships in Singapore, finding that increases in temperature and precipitation increase dengue risk, as well as El Niño conditions[26,30]. However, we also found evidence of non-linearity in the temperature-dengue relationship, with decreased risk at high maximum temperatures. A previous study by Xu and colleagues found that absolute humidity was a better predictor for dengue incidence than other climatic variables, due to the stability of its relationship with dengue incidence during different subperiods of serotype circulation[23]. Contrastingly, in this study, we accounted jointly for the impact of serotype circulation and local climatic indicators on dengue risk, and therefore were able to estimate the marginal effects of each climatic indicator without confounding from concurrent serotype dynamics. As a result of this, we found stronger evidence for a role of temperature and rainfall in transmission, with little additional benefit of humidity information.

El Niño is thought to affect dengue transmission through changes in local weather conditions. We included covariates to capture both El Niño and local weather conditions within the same model structure, following the logic that the full effect of ENSO on transmission is unlikely to be fully captured by the weather covariates included in the model. For instance, ENSO may affect humidity, which was not explicitly included in the model, or temperature and precipitation metrics other than those included in the model, potentially affecting dengue season length or timing, or the spatial spread of dengue in Singapore.

We adapted our model into a proposed early warning framework, generating accurate forecasts at operationally useful lead times[30]. We used a rigorous time series cross-validation methodology to realistically evaluate model performance for early warning. This retrospective statistical validation is a necessary step in the construction of a forecasting model for early warning. When a model is implemented operationally, forecasts may then be used for decision-making, which can directly impact dengue transmission and complicate the evaluation of forecast accuracy.

Incorporating serotype dynamics within our forecasting framework helped to account for interannual variability in dengue transmission, improved dengue case forecasts at shorter lead times (up to 4 weeks ahead) and increased outbreak detection accuracy at all lead times. By comparing our full *climate and serotype* model with models only containing *climate* or *serotype* information, we were able to identify time periods where particular covariates were improving predictions. For instance, including serotype information helped to predict decreases following peaks in 2013 and 2014, as well as the peak in 2019. Contrastingly, including climate information helped to predict peak intensity and timing in 2016, during an El Niño event. It should also be noted that while our *climate and serotype* and *climate only* models perform equally well when forecasting dengue cases at longer lead times (6–8 weeks ahead), this is largely due to the flexible random effects incorporated in the modelling framework. When generating probabilistic forecasts, the covariate models estimate yearly random effects based on data available for that year up until the forecast target week. We conducted sensitivity analysis around this, running these

models without a yearly random effect. In this case, the *climate only* model performs worse than the *climate and serotype* model at all forecast horizons (Supplementary Figs. 5 and 6). While this demonstrates that including serotype information is helping to predict interannual variability, it also highlights the power of a flexible random effects structure to account for unmeasured variation and improve seasonal forecasts. As such, this framework could also be applied to other dengue-endemic settings without regular serotype surveillance data, albeit with potentially lower predictive power.

This analysis was co-designed with the NEA to address key questions around local dengue transmission. The proposed model offers a more interpretable forecasting framework than current machine learning based forecasts, such as a LASSO framework, as the strength of the association between climatic predictors and dengue cases can vary by forecast horizon and cannot be translated directly into dengue risk[28]. We compared our proposed INLA model framework with an updated version of a previously published LASSO model developed for operational use by the NEA. We found that both models showed good predictive ability but that they differed in accuracy over time, suggesting an ensemble approach may be beneficial. We also demonstrated the potential utility of our framework to conduct scenario analysis, generating dengue predictions according to observed climate and serotype dynamics to estimate the impact of novel dengue interventions. We explored the potential impact of the *Wolbachia-Aedes* mosquito suppression strategy currently being rolled out in Singapore, estimating that the expansion in *Wolbachia* releases in mid-2022 averted around 28% of cases in 2023, according to the median case count expected in a counterfactual scenario without this expansion in releases. Further extensions of this work could include combining the proposed INLA model with other forecasting approaches in an ensemble modelling framework. Additionally, this model could also be used to generate probabilistic forecasts under different scenarios at the beginning of a dengue season to understand potential outbreak risks and inform public health planning (e.g., comparing forecasts with or without a switch in dominant serotype or under different possible Niño 3.4 SSTAs).

Despite this, there are several limitations to this study. Our *climate and serotype* model struggled to predict particularly large peak weekly cases in 2020 and 2022. We included serotype dynamics using the time since a switch in dominant serotype as a proxy for population immunity, which potentially underestimates cases during periods where a non-dominant serotype is growing in prevalence. In Singapore, seroprevalence surveys are conducted every 5 years, limiting the utility of this raw data as an input into a short-term statistical forecasting framework; however, it is possible that modelled estimates of population susceptibility could further improve predictive power. We were also unable to include serotype-specific dynamics due to the low number of switches in dominant serotype, even with a 17 year period of virus surveillance. This could be important to include as certain serotypes are associated with greater clinical severity and secondary infections are thought to be associated with increased severity as a result of antibody-dependent enhancement[37,38]. Similarly, we only consider switches in antigenic serotype and do not consider changes in genotype prevalence or genetic distance between serotypes, which are also hypothesised to increase outbreak risk[39,40]. We did not incorporate vector density into the model, as it mediates the relationship between climatic conditions and dengue transmission. We did not have access to data on routine vector control efforts, therefore, any effects of interventions would be accounted for via yearly random effects. However, it should be noted that vector control is stringently maintained within Singapore and vector density has been low for several decades. Finally, we conducted model evaluation with forecast skill metrics and ROC analysis. In an ideal implementation scenario, forecast outputs such as outbreak warning alerts would be tied to specific control interventions or public health actions[29]. This would allow for

the cost of a false alarm or missed event to be calculated and enable a cost-effectiveness analysis of the early warning system as a whole.

Our analysis disentangles the role of climate and serotype dynamics in driving dengue outbreaks over a 23-year period in Singapore, capitalising on a rich dataset of epidemiological, weather station and virus surveillance data. We translate these findings into an early warning framework able to forecast dengue cases and generate outbreak alert predictions with 0–8 weeks lead time. In this study, we integrate explanatory and predictive modelling approaches with a view that understanding the key causal relationships underlying transmission allows for the construction of forecasting models that are more generalisable[41]. This can also result in greater interpretability, allowing for clearer communication of forecasts to policymakers and a more intuitive understanding of how transmission may vary under future large-scale changes, such as climate change. Climate-driven early warning systems will become increasingly important adaptation measures as climate change alters the geographical range of dengue transmission and leads to greater climatic extremes. We demonstrate the additional value of viral surveillance in improving forecast accuracy, and particularly in addressing the challenge of predicting dengue outbreak years.

## Methods
### Data
In Singapore, dengue case reporting by clinicians and laboratories is mandatory. Weekly laboratory-confirmed cases from 1 January 2000 to 31 December 2022 were provided by the Ministry of Health, Singapore[42]. Laboratory confirmation is performed through antigen detection of nonstructural protein 1 (NS1) or detection of viral RNA by polymerase chain reaction (PCR) in the first five days of illness, or serological detection of immunoglobulin M after five days of illness[7]. We define an outbreak using a seasonal moving 75th percentile threshold. For a given month and year, we defined an outbreak threshold of the 75th percentile of weekly cases in that month using all years up to (but not including) the given year. This identifies the same outbreak periods as the endemic channel threshold used operationally in Singapore, but has several added benefits. A percentile threshold is simpler to calculate and adjusts on a rolling basis rather than year-on-year. Additionally, by incorporating the seasonal patterns underlying dengue transmission, this definition allows outbreak periods to be defined earlier than with a fixed, non-seasonal threshold (Supplementary Fig. 1).

Since 2006, the NEA's Environmental Health Institute has serotyped a subset of dengue samples using RT-PCR as part of a virus surveillance programme[8]. In our dataset, for years where serotype information is available, ~27% of reported dengue cases are serotyped. Weekly DENV1-4 frequencies and total number of serotyped samples were provided by the Ministry of Health[42]. We calculated a smoothed proportion for DENV1-4 for each week with a generalised additive model multinomial logistic regression using *mgcv 1.8.36*[43]. To smooth the serotype data, we only used data up to and including the week of interest to enable resulting models to be useful in a forward-looking early warning framework. We then use these smoothed DENV proportions to classify the dominant serotype for each week. We defined a switch event as occurring where the current dominant serotype is different from the dominant serotype in the previous week, identifying 4 switch events in our dataset. We then defined the time since a switch in dominant serotype as the number of weeks since the most recent switch event.

Midyear population size estimates were obtained from the Singapore Department of Statistics, which included both local and foreigner populations.

Weekly maximum, minimum and mean temperature (°C), absolute (g/m³) and relative humidity (%), and precipitation (mm) were provided by the NEA. Daily precipitation (mm) was also provided and

used to calculate: the number of days without rain per week (calculated as $\sum P_d = 0$, where $P_d$ represents rainfall on a given day); number of days with heavy rain per week (calculated as $P_d \geq 40$); number of days with moderate to heavy rain per week (calculated as $P_d \geq 20$); and number of days with consecutive rainfall (i.e., a count of consecutive days where $P_d \geq 1$). Thresholds were chosen based on exploratory analysis of daily rainfall data and broadly align with 90th and 97.5th percentiles. Weekly Niño 3.4 SSTA was obtained from the National Oceanic and Atmospheric Administration[44]. This index measures the ENSO, interannual fluctuations in the oceanic and atmospheric temperature around the Pacific Ocean. The index is commonly used to define El Niño events (unusually warm) and La Niña events (unusually cool). Sea surface temperature anomalies are calculated by subtracting the observed sea surface temperatures from a historical mean, calculated for the period 1981–2010. El Niño and La Niña events are typically defined by a sea surface temperature anomaly of +/− 0.5 °C for over 6 months.

## Model framework

We used a Bayesian hierarchical mixed-effects model to produce probabilistic predictions of weekly dengue incidence. Inference was performed using integrated nested Laplace approximation in INLA 23.04.24[45]. Weekly dengue counts ($y_t$) were assumed to follow a negative binomial distribution to account for overdispersion in the data, with a mean $\mu_t$ and overdispersion parameter $\kappa$.

$$y_t \sim NegBin(\mu_t, \kappa) \tag{1}$$

$$\log(\mu_t) = \log(P_{a[t]}) + \alpha + \sum \beta_k X_{k,t} + \gamma_{a[t]} + \delta_{w[t]} \tag{2}$$

Here $\log(\mu_t)$ is the linear predictor, where $\log(P_{a[t]})$ is a population offset with population per 100,000 by year and $\alpha$ is the model intercept. $\sum \beta_k X_{k,t}$ is a vector of $k$ climate and/or serotype covariates, $\gamma_{a[t]}$ is a yearly random effect and $\delta_{w[t]}$ is a weekly random effect. We use the weekly random effect to account for seasonality and seasonal autocorrelation, while the yearly random effect accounts for any unexplained interannual variation in the data, for example as a result of vector control efforts or COVID-19 restrictions[46]. Details on model prior specifications and hyperparameters are available in the Supplementary Materials.

## Model selection

We calculated Pearson's rank correlation index to assess correlation between variables using *corrplot 0.92* and considered $r \geq 0.5$ as indicative of high correlation. We also calculated the variance inflation factor to assess multicollinearity, considering $VIF \geq 5$ as evidence of high collinearity. Based on this, we excluded relative humidity from further analysis due to high correlation with temperature variables and rainfall.

We tested all remaining variables with a 0, 4, 8, 12 and 16 week lag. For temperature, humidity and Niño variables, we tested 1, 4, 8 and 12 week running averages, while for precipitation variables we tested 1, 4, 8 and 12 week running totals. Finally, we also tested non-linear formulations of temperature, precipitation and Niño 3.4 variables. We explored the best combinations of different classes of covariate (temperature, precipitation, humidity and Niño), conducting model selection in a forward stepwise manner, comparing models of increasing complexity (Supplementary Table 2). Covariate models were compared to a *random effects only* model including weekly and yearly random effects ($\gamma_{a[t]} + \delta_{w[t]}$). Overall, 505 model formulations of climatic and serotype covariates were tested. We used various model adequacy criteria including: the widely applicable information criteria (WAIC) and deviance information criteria (DIC). WAIC and DIC are

metrics which aim to maximise model fit while also penalising model complexity, with lower scores indicating a more parsimonious model.

Once we had selected the best performing climate model, we tested the inclusion of serotype variables including: dengue serotype proportions (individually and in combination, numeric variables), dengue serotype growth rates (individually and in combination, numeric variables), yearly or weekly dominant serotype (factor variable with four levels), serotype switch event (binary variable), time since serotype switch (numeric).

We quantified the contribution of model covariates in reducing unexplained variation in dengue incidence by comparing the estimated precision of the yearly random effect $\gamma_{a[t]}$ between the covariate model and the baseline model. We assessed this using the ratio of marginal variances calculated as

$$\frac{1}{\tau_{model}} \Big/ \frac{1}{\tau_{baseline}}$$

Where $\tau$ is the estimated marginal precision for the yearly random effect. Here, a ratio near 1 suggests that the covariates cannot help explain the interannual variation accounted for by the random effects in the baseline model. A ratio less than 1 suggests that covariates are capturing some of the unexplained interannual variation.

## Model evaluation

We evaluated the performance of four candidate models: a *climate and serotype* model, a *climate only* model, a *serotype only* model, a *seasonal baseline* model. We conducted model evaluation using time series cross-validation methodology to produce posterior predictions, and comparing observed and predicted outcomes to evaluate predictive performance. This is an appropriate cross-validation design when conducting statistical validation of forecasting models, as it preserves the time order of the underlying data, i.e., only data prior to an observation occurring is used to generate the prediction. To do this, we refit the model for each week in the dataset from January 2009 to December 2022 using an expanding window approach[35]. Data for the first eight years was used solely for training. Then, for each target week $t$ in the dataset, we trained the model on data until week $t − 1$ and then simulated a posterior predictive distribution for dengue cases in week $t$, using climatic data up until time $t$. Serotype covariates are constructed only using data until $t − 1$ for each time point $t$ as serotype frequencies are linked to dengue case counts. The posterior predictive distribution was simulated using 1000 samples from the posterior distribution of model parameters and hyperparameters. Note that all our final models (except the *seasonal baseline*) included a yearly random effect, $\gamma_{a[t]}$, which is estimated using only using data available until week $t$ for the year of the target week.

We also calculated posterior predictive probabilities of exceeding the outbreak threshold. This both evaluates the model's ability to distinguish between outbreak and non-outbreak periods, and provides an operationally useful model output for decision makers. For instance, it may be easier to tie specific response measures or public health decisions to a probability of exceeding an operationally meaningful threshold than to the full probabilistic forecast. Here, we calculated outbreak thresholds for each week based on the seasonal moving 75th percentile of cases (Supplementary Fig. 1). For each week $t$, we then calculated the proportion of posterior predictive samples that were greater than the threshold value.

Forecasts were scored using the *scoringutils 1.2.1* package[47]. We aimed to evaluate both how successful models were at predicting dengue case incidence and detecting outbreak weeks. Dengue case forecasts were scored using the CRPS; this is a proper scoring rule which can be considered a generalisation of mean absolute error that takes into account the entire predictive distribution[48]. Smaller CRPS values indicate a better forecast, and the metric penalises both under-

and over-prediction. Sharper forecasts (where predictions are concentrated in a narrower range) will also score better. We then calculated the CRPSS, which is calculated as $1 - \frac{CRPS_{model}}{CRPS_{baseline}}$ and measures the percentage improvement of the considered model over a baseline model. A value of 1 indicates that the model has perfect skill compared to the baseline, 0 indicates the model is equivalent to the baseline and a negative value indicates that the model is worse than the baseline.

We also assessed model calibration by calculating interval coverage at the 50% and 95% levels. Interval coverage measures the proportion of observed values falling in a given prediction interval range. For a given prediction interval, a perfectly calibrated model would have interval coverage equal to the nominal prediction interval (that is, 95% of observations falling within the 95% prediction interval). We also calculated bias $B$, measuring a model's tendency to over or under-predict. This was calculated for a data point of weekly cases $y_t$ such that:

$$B(P, y_t) = 1 - (P(y_t) + P(y_t - 1)) \qquad (3)$$

Where $P(y_t)$ is the predicted probability mass for all outcomes smaller or equal to $y_t$.

To score the models' ability to forecast future outbreaks, we calculated the Brier score comparing the posterior predictive probability of exceeding the outbreak threshold with observed outbreak weeks. The Brier score is a proper scoring rule for binary outcomes computed as the mean square error between the probabilistic prediction and the true outcome, where smaller values indicate better forecasts[49]. Finally, we used ROC analysis to determine the optimum threshold for issuing outbreak alerts, balancing hit rate with false alarm rate, using the *pROC package 1.18.4*[50,51]. Here, hit rate (or sensitivity) is defined as the proportion of events (outbreak weeks) that were correctly predicted. False alarm rate (or 1−specificity) is defined as the proportion of weeks without an outbreak where an outbreak was predicted to occur. We generated ROC curves for each model, which show hit rate against false alarm rate at different outbreak alert thresholds and calculated the area under the ROC curve (AUC). The AUC is a measure of model performance in classifying outbreak and non-outbreak weeks, with higher values indicating a better classification[52]. We selected outbreak alert thresholds by choosing the point closest to the top left of the ROC plot (representing perfect sensitivity or specificity).

### Adapting the model for early warning

To adapt our model framework for use in an early warning scenario, we produced and evaluated dengue case forecasts at 2, 4, 6 and 8 week ahead forecast horizons using the same expanding window time series cross-validation approach. To do this, we trained the model using the final selected covariates on data available until $t - h$ (where $h$ represents the forecast horizon). Then, using these estimated model parameters, we predicted dengue cases at week $t$ using the best approximation of each covariate in the final models available at the point of forecast. For instance, in our model selection process, we found that a 12 week running average of maximum temperature (°C) with no lag was the best performing temperature indicator. To produce a forecast for time point $t$ at a 4 week ahead time horizon, we used an 8 week running average of maximum temperature (°C) available at the forecast issue date ($t - 4$). Similarly, we found the best performing rainfall covariate was a 12 week total of days without rain. To approximate this for a 4 week ahead time horizon we used an 8 week total of days without rain, at point $t - 4$, scaled up by a factor of 1.5. Full details of the variables used for prediction at each time horizon are available in Supplementary Table 5 and a schematic showing the cross-validation design for different forecast horizons is shown in Supplementary Fig. 3. This approach allows us to preserve the key relationships between climate and serotype covariates, and dengue

cases that we estimate in full model fitting and then use the best climate data available at different lead times to generate forecasts for early warning.

### Counterfactual scenario analysis

We conducted counterfactual scenario analysis to estimate the impact of the *Wolbachia-Aedes* suppression programme on dengue cases in the 2023 dengue season. Here, we fit our final *climate and serotype* model using data until 26th June 2022. We then generated predictions of expected dengue cases using observed climate and serotype data until the end of 2023. We estimated a posterior distribution for the predictions, which includes uncertainty associated with the linear predictor (including fixed and random effects) but not observation error from the overdispersion parameter. We calculated the estimated cases averted by subtracting the observed cases from the median predicted counterfactual cases and the percentage of the estimated cases averted as

$$\frac{Cases\ averted}{Median\ predicted\ counterfactual\ cases} \times 100.$$

### Ethical approval

The London School of Hygiene and Tropical Medicine Ethics Committee waived ethical approval for this work. This study was exempted from national ethical review in Singapore as it is not considered human biological research.

### Reporting summary

Further information on research design is available in the Nature Portfolio Reporting Summary linked to this article.

## Data availability

Surveillance and climatic data were shared by the National Environment Agency of Singapore. Dengue case data are publicly available and can be found in weekly infectious disease bulletins at: https://www.cda.gov.sg/resources/weekly-infectious-diseases-bulletin-2025/. Processed data used in this analysis are available at: https://github.com/EmilieFinch/dengue-singapore with the exception of serotype prevalences shown in Fig. 1B as these are not publicly available due to institutional policy. Data requests may be submitted to Chia-chen Chang at chang_chia-chen@nea.gov.sg. Access may be granted under data use agreements for a specified research scope. Requests are typically responded to within 1–2 weeks.

## Code availability

All code used for this analysis are available at: https://github.com/EmilieFinch/dengue-singapore. Repository reference: Finch E, Chang CC, Kucharski A, Sim S, Ng LC, Lowe R. Climate variation and serotype competition drive dengue outbreak dynamics in Singapore. dengue-singapore. https://doi.org/10.5281/zenodo.17250318 2025.

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

## Acknowledgements

E.F. was supported by the Medical Research Council (MR/N013638/1); R.L. acknowledges support from the Wellcome Trust (IDExtremes 226069/Z/22/Z), the EU's Horizon Europe research and innovation programme (E4Warning; grant agreement 101086640) and a Royal Society Dorothy Hodgkin Fellowship; A.J.K. was supported by Wellcome Trust (206250/Z/17/Z); S.S., C.-C.C., and L.C.N. are employees of the National Environment Agency, Singapore.

## Author contributions

Conceptualisation: E.F., R.L., S.S., L.C.N. Methodology: E.F., R.L., A.J.K. Data curation: C.C.C., S.S., L.C.N., E.F. Investigation: E.F., R.L. Supervision: R.L., A.J.K. Data analysis: E.F., C.C.C. Figures: E.F. Writing–original draft: E.F., R.L., A.J.K. Writing–review and editing: all authors.

## Competing interests

The authors declare no competing interests.
