## [Peer Review File · Nature Communications]

Climate variation and serotype competition drive dengue outbreak dynamics in Singapore

Corresponding Author: Dr Emilie Finch

Version 0:

Reviewer comments:

Reviewer #1

(Remarks to the Author)

See attachment.

(Remarks on code availability)

Response to reviewer comments

1st October 2025

This study integrates over two decades of dengue surveillance data and applies a Bayesian hierarchical modeling framework to quantify the impact of climatic variation and serotype competition on dengue dynamics in Singapore. The methodological rigor, clarity in presentation, and the practical implications (e.g., forecasting and evaluating Wolbachia intervention) are noteworthy and commendable. After careful reviewing of this manuscript, I believe there are some minor comments that need to be changed before publication: (1) The study should discuss the generalizability of the model to other dengue-endemic settings, particularly those lacking serotype surveillance data.

Thank you for this positive feedback on our manuscript. We discuss the applicability of this framework for settings without serotype surveillance data in line 326. In this section we refer to a sensitivity analysis where we ran our forecasting models without a yearly random effect (Supplementary Figures 5-6). We state here that “While this demonstrates that including serotype information is helping to predict interannual variability, it also highlights the power of a flexible random effects structure to account for unmeasured variation and improve seasonal forecasts, particularly for settings without regular virus surveillance”. We’ve added clarity here by adding a sentence stating that “As such, this framework could also be applied to other dengue-endemic settings without regular serotype surveillance data, albeit with potentially lower predictive power”.

(2) The manuscript states that only approximately 27% of cases were serotyped. It is necessary to discuss the potential bias introduced by untyped cases (e.g., whether the missingness is at random).

Thank you for this comment. In Singapore, the laboratory based virus surveillance program at the Environmental Health Institute collects samples from any suspected DENV cases in a network of hospitals and general practitioners. Any positive DENV samples are then serotyped and, in addition to this, any DENV positive samples sent from commercial laboratories to the EHI are also serotyped. As such, the potential bias should be minimal.

(3) Using the time since serotype switch as a proxy for immunity may be overly simplistic. It is recommended to discuss whether the unavailability of other potential immunity indicators (e.g., seroprevalence) affects the conclusions.

Thank you for this comment. During model development we tested more complex indicators, including continuous variables for the proportion of each serotype present and the growth rate of each serotype (Supplementary Table 1). We found that these were outperformed by a simpler ‘time since a switch in the dominant serotype’ variable. We hypothesise this is because there were only four serotype switch events in our dataset and few years with substantial transmission of DENV-3 and DENV-4, and

so we did not have the statistical power to resolve the contribution of each serotype to overall risk. However, it is possible that this indicator results in underestimates of cases during periods where a non-dominant serotype is growing in prevalence. While we agree that incorporating seroprevalence data would be interesting and potentially beneficial, in our study setting, Singapore, seroprevalence surveys are conducted every 5 years, limiting the utility of this data in a statistical forecasting framework predicting 8 weeks ahead. We have commented further on this question in line 353 where we now state:

“We included serotype dynamics using the time since a switch in dominant serotype as a proxy for population immunity, which potentially underestimates cases during periods where a non-dominant serotype is growing in prevalence. In Singapore, seroprevalence surveys are conducted every 5 years, limiting the utility of this raw data as an input into a short-term statistical forecasting framework; however, it is possible that modelled estimates of population susceptibility could further improve predictive power”.

(4) ENSO and local climate collinearity: Although the model included both ENSO and local climate variables, their independence should be verified (e.g., through VIF values).

Thank you for this comment, in our manuscript we did assess the degree of collinearity between ENSO and local climate variables using a VIF threshold of 5 as evidence of high collinearity. Based on this we kept both ENSO and local climate variables in the model (but excluded relative humidity due to high correlation with temperature and rainfall variables).

This is mentioned on line 454 where we state “We calculated Pearson’s rank correlation index to assess correlation between variables using *corrplot* 0.92 and considered $r \geq 0.5$ as indicative of high correlation. We also calculated the variance inflation factor to assess multicollinearity, considering $VIF \geq 5$ as evidence of high collinearity. Based on this, we excluded relative humidity from further analysis due to high correlation with temperature variables and rainfall.”

(5) Role of random effects: It is recommended to clarify in the results what unmeasured factors are captured by the yearly random effects (e.g., vector control intensity).

Thank you for this comment. We have added a comment on this to line 111 which now reads “We used a negative binomial likelihood and incorporated weekly random effects to capture seasonality and yearly random effects to account for unexplained interannual variation in dengue risk, for instance due to changes in vector control intensity (Methods).”

(6) All abbreviations should be spelled out at first mention (e.g., SSTA). Ensure that all supplementary figures are referenced in the main text, and key results are indicated.

Thank you for this feedback, we have added full phrases for all mentioned abbreviations. All supplementary figures are referenced in the main text with their key results described.

This study integrates over two decades of dengue surveillance data and applies a Bayesian hierarchical modeling framework to quantify the impact of climatic variation and serotype competition on dengue dynamics in Singapore. The methodological rigor, clarity in presentation, and the practical implications (e.g., forecasting and evaluating Wolbachia intervention) are noteworthy and commendable. After careful reviewing of this manuscript, I believe there are some minor comments that need to be changed before publication:

- (1) The study should discuss the generalizability of the model to other dengue-endemic settings, particularly those lacking serotype surveillance data.
- (2) The manuscript states that only approximately 27% of cases were serotyped. It is necessary to discuss the potential bias introduced by untyped cases (e.g., whether the missingness is at random).
- (3) Using the time since serotype switch as a proxy for immunity may be overly simplistic. It is recommended to discuss whether the unavailability of other potential immunity indicators (e.g., seroprevalence) affects the conclusions.
- (4) ENSO and local climate collinearity: Although the model included both ENSO and local climate variables, their independence should be verified (e.g., through VIF values).
- (5) Role of random effects: It is recommended to clarify in the results what unmeasured factors are captured by the yearly random effects (e.g., vector control intensity).
- (6) All abbreviations should be spelled out at first mention (e.g., SSTA). Ensure that all supplementary figures are referenced in the main text, and key results are indicated.